# Application of Machine Learning for Prediction and Process Optimization—Case Study of Blush Defect in Plastic Injection Molding

**Alireza Mollaei Ardestani** [1] , **Ghasem Azamirad** [2,*], **Yasin Shokrollahi** [3] , **Matteo Calaon** [1] , **Jesper Henri Hattel** [1] , **Murat Kulahci** [4] , **Roya Soltani** [5] and **Guido Tosello** [1]

1   Department of Civil and Mechanical Engineering, Technical University of Denmark, 2800 Kgs. Lyngby, Denmark
2   Department of Mechanical Engineering, Yazd University, Yazd P.O. Box 8915818411, Iran
3   Department of Mechanical Engineering, Amirkabir University of Technology, Tehran P.O. Box 15875-4413, Iran
4   Department of Applied Mathematics and Computer Science, Technical University of Denmark, 2800 Kgs. Lyngby, Denmark
5   Yazd Poolika Co., Yazd Industrial Zone, Yazd P.O. Box 8947183963, Iran
*   Correspondence: azamirad@yazd.ac.ir

**Abstract:** Injection molding is one of the most important processes for the mass production of plastic parts. In recent years, many researchers have focused on predicting the occurrence and intensity of defects in injected molded parts, as well as the optimization of process parameters to avoid such defects. One of the most frequent defects of manufactured parts is blush, which usually occurs around the gate location. In this study, to identify the effective parameters on blush formation, eight design parameters with effect probability on the influence of this defect have been investigated. Using a combination of design of experiments (DOE), finite element analysis (FEA), and ANOVA, the most significant parameters have been identified (runner diameter, holding pressure, flow rate, and melt temperature). Furthermore, to provide an efficient predictive model, machine learning methods such as basic artificial neural networks, their combination with genetic algorithms, and particle swarm optimization have been applied and their performance analyzed. It was found that the basic artificial neural network (ANN), with an average accuracy error of 1.3%, provides the closest predictions to the FEA results. Additionally, the process parameters were optimized using ANOVA and a genetic algorithm, which resulted in a significant reduction in the blush defect area.

**Keywords:** plastic injection molding; design of experiments; machine learning; digital twin; process optimization

## 1. Introduction

With the widespread use of algorithms in the engineering sciences, lots of additional costs associated with time-consuming and expensive tests have been eliminated from the product design and production development cycle. Today's modeling, prediction, and optimization methods have extremely reduced the need to use traditional experimental trials and measurements for product and process improvement. These techniques include statistical methods such as Analysis of Variances (ANOVA), machine learning methods such as Artificial Neural Networks (ANNs), as well as optimization methods using meta-heuristic algorithms. Nowadays, the use of Finite Element Analysis (FEA) methods combined with modern optimization methods has helped manufacturers select the optimal levels of input parameters and achieve the highest quality products [1,2]. Due to the complex behavior of polymers, particularly when processed by injection molding, many parameters can affect the product quality. Hence, monitoring and controlling each of these parameters as well as their interactions is vital to prevent injection defects. For example, a fundamental quality

of injection molded parts is based on the reduction of warpage. Among statistical methods, Genetic Algorithm (GA) is known as a popular technique for reducing warpage [3], as well as other defects. Additionally, ANNs have been widely used to predict the shrinkage values of the examined parts [4]. Wang et al. [5], by taking an ANN approach, found that the shrinkage of the manufactured part has an inverse relationship with process parameters such as packing time, injection pressure, holding pressure, and melt temperature, and a direct relationship with cooling time and mold temperature. Altan et al. [6] conducted a study to reduce the shrinkage of a manufactured part. They studied parts composed of polypropylene and polystyrene. After the design of experiments based on the Taguchi method, an ANOVA was performed. As a result, optimal levels for design parameters were obtained, and by employing these optimal parameters in the simulation, the shrinkage of the parts was reduced significantly. An ANN was also trained to predict response values, and it was found that this kind of network has a good ability for this purpose. In a study conducted by Chen et al. [7], it was found that the parameters of holding pressure and melt temperature are the most effective factors in the warpage of the parts. A model was also presented that demonstrated the ability to predict parts warpage in both simulations and experimental tests with relatively high accuracy. It is also possible to use several optimization methods, such as Taguchi method, ANN, GA, Response Surface Methodology (RSM), Particle Swarm Optimization (PSO), etc. simultaneously to optimize the parameters involved in the part's warpage. Finally, according to the prediction error and the number of iterations performed, the best optimization method can be selected [8–11]. The Taguchi optimization method has been used to reduce the warpage of the parts and improve their strength. Consequently, the holding pressure was found to be the most significant factor in the warpage of the parts [12]. Similar research has been conducted using the Taguchi, Non-dominated Sorting Genetic Algorithm (NSGA-II), and RSM methods to optimize warpage, shrinkage, and residual stress by Li et al. [13]. Reduction of warpage by considering mold filling as a constraint using the RBF (Radial Basis Function) method was achieved in [14]. The RBF method was also used to perform a multi-objective optimization to reduce production time, clamping force, and warpage in the study by Kitayama et al. [15]. ANNs and GA were used to study shrinkage. Researchers found a relationship between design variables and the target parameter, which showed a great ability of ANN to predict the results [16]. In two separate studies, Kitayama et al. arranged two multi-objective optimizations that both used the RBF method to improve the weld lines. In one of these studies, the weld lines were optimized while reducing the clamping force. In a second study, the goal of the research was to reduce cycle time and weld lines [17,18]. Other than the determination of the optimal levels for the factors related to the process parameters, meta-heuristic algorithms have been also used to determine the optimal values for the composition ratio of polymers to reduce the maximum shrinkage in the molded products [19]. Xu et al. used the Taguchi method, ANN, multi-objective PSO algorithm, and a combination of ANN with PSO to attain the optimal level for design parameters to reduce the part weight, shrinkage, and flash size [20]. Along with the shrinkage of the produced part, other parameters such as clamping force have been inspected simultaneously in a multi-objective optimization using the RBF method [21]. To reduce the part shrinkage, the cycle time, and injection time at the same time, a multi-objective optimization using an ANN-based program was carried out [22]. Some studies have examined other types of defects. For example, the research by Tabi et al. [23] aimed at improving the needle-shaped defects around the gate location. The role of residual stress in the occurrence of cracks in parts [24], and the use of RSM for cutting down the average shear rate around the gate [25] were also investigated. Furthermore, improving the optical properties of polymer optics as a result of reducing residual stress in injection molded plastic lenses was the subject of another study [26]. Li et al. employed Kriging and NSGA-II methods to investigate the effect of runner diameter and process parameters on the quality, cost, and production efficiency [27]. The Taguchi method has been used to investigate the effect of process parameters on the mechanical properties of parts produced with recycled plastic. It was revealed that injection time with

40.5% and material temperature with 43.3% of impact are the most important factors in warpage and yield stress of the produced part, respectively [28]. Additionally, Martowibowo et al. [29] used a GA, with a prediction error of approximately 1%, and could find the optimal values of process parameters in such a way that the production time of the part was significantly reduced. In other studies, FEM and conventional optimization methods were used to decrease the cycle time to a minimum by providing an optimal design for the cooling system [15,30]. Eladl et al. [31] studied the effect of process parameters on the formation of flash defect. During the study, polypropylene (PP) and acrylonitrile butadiene styrene (ABS) were investigated. The outputs of the study were part mass, flow length, and flash formation. After employing DOE and statistical analysis, it was shown the injection speed and packing pressure were the most influential factors on the flash area for both materials. Regi et al. [32] used a different technique to investigate the flow propagation in the molds. They used a transparent window on one of the walls of the mold to visually observe the flow of the material. The objective of the study was to compare FEA simulation results with experiments with focus on flow hesitation. A high-speed camera was used to record the mold filling phase with two different materials of PP and ABS. After employing DOE and ANOVA, the results demonstrated that flow progression and hesitation are dependent on wall thickness, injection velocity, and material type. Loaldi et al. [33], using the same techniques (DOE and FEA) focused on the experimental validation of the injection molding simulation of microparts, also with a focus on flash formation. Results confirmed that higher values of holding pressure, injection speed, mold temperature, melt temperature generate larger flash areas. The trends were correctly predicted by the FEA flow simulations. Chen et al. [34] employed the Taguchi, ANOVA, backpropagation ANN, GA, and Davidson-Fletcher-Powell methods to improve the product quality. As a result of this optimization, as well as increasing demands in terms of products quality, issues such as waste, number of defective parts, need for inspection during production, need for recycling, and production time were also decreased. By using machine learning methods, Finkeldey et al. [35] were able to accurately predict the weight and thickness of the manufactured part. Mehat et al. [36] used process parameters optimization instead of adding additives to improve the mechanical properties in a part produced with recycled plastic. Clearly, considerable research efforts have been focused recently to optimize injection molding parts characteristics and minimize several types of defects. However, very limited research has investigated the factors affecting the blush defect. At least, one study can be indicated to have addressed the influence of process parameters on blush: results by Llado et al. [37] indicated that injection flow rate and melt temperature are the most effective parameters. Blush defect not only deteriorates the appearance of the part but also reduces the lifetime and strength of the product. Therefore, due to its importance and the limited research performed so far, there is a clear need for further investigation on the parameters affecting the incidence and exacerbation of this defect. Despite the importance of preventing the incidence of this defect in plastic injection, so far, only a few studies have investigated its causes and impacts.

The present research has inspected the effects of eight injection molding factors (flow rate, melt temperature, holding pressure, mold temperature, runner diameter, gate diameter, gate angle, included angle) on the blush. The novelty of the present study is the relatively number of investigated parameters, including both process and design factors. By examining a large variety of parameters, the study results in a comprehensive view of the causes of this defect. What is more, contrary to previous research, the interaction effects of design parameters on the incidence of the defect have also been studied in the current study. In addition, for the first time, different types of ANNs have been used to predict the values of blush defect, and a GA is utilized to optimize the levels of effective factors to achieve the lowest probable blush defect.

The research method of this study began with the creation of a CAD model of the injection molded component (plastic bushing). Then, as screening step, the fractional factorial DOE with two levels and the eight design parameters has been conducted. After

performing FEA, the results were included in an ANOVA routine to identify effective and ineffective factors. Thereafter, for a more detailed study of the effect of the most relevant parameters, a second design of experiment of CCD (Central Composite Design) type has been performed with five levels on the effective factors.

## 2. Materials and Methods

### 2.1. Modeling

Blush is a visual defect that occurs as white halos, usually around the gate location. The part under study was a size bushing produced with an injection mold equipped with two cavities. Bushings are a kind a fitting in piping that can be used to connect two pipes together, change the pipeline and flow direction, derive new pipe branches, and to blind a branch. Polyvinyl chloride (PVC) fittings are commonly used in industry due to their ease of use, durability, and cost effectiveness. The geometry of the bushing and injection mold runners is shown in Figure 1. The gate is of round type, and the length of the sprue is 76 mm with a conic angle of 3°. All the geometrical dimensions to reproduce the CAD model are included in Figure 1.

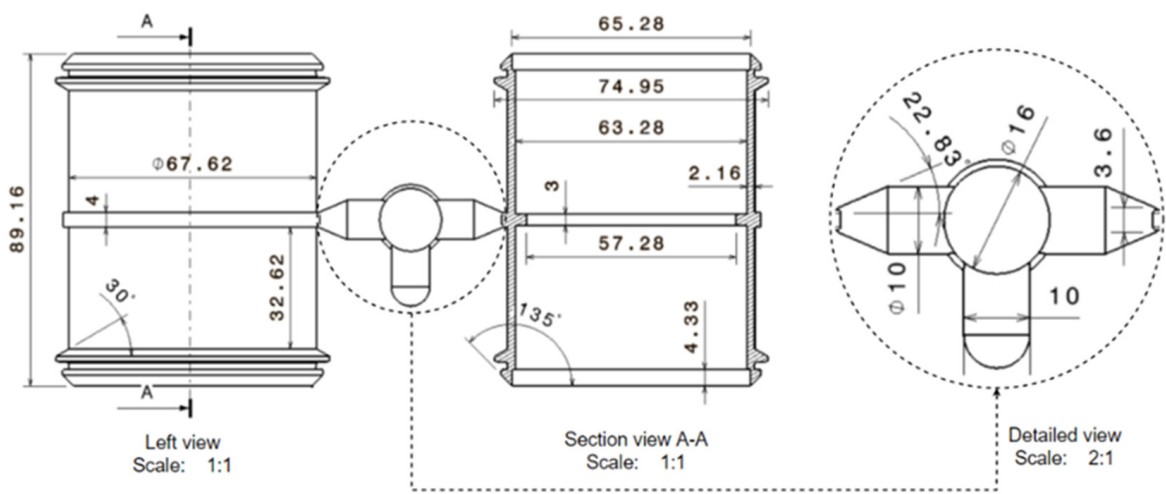

**Figure 1.** Bushing geometry and dimensions of runner system of the original mold.

The material used in the study is a commercially available grade of polyvinyl chloride (PVC) produced by Solvay ET CIE (Brussels, Belgium) under the commercial name Benvic IR705. This type of PVC is widely used in the production of pipes and fittings. Some of its mechanical, thermal, and rheological properties are presented in Table 1 [34]. Figure 2 shows the PVT data and viscosity characteristics (pressure, specific volume, and temperature) of the material based on the Autodesk Moldflow software database.

**Table 1.** Properties of Benvic IR705 [34].

| Parameter | Value |
|---|---|
| Maximum allowable shear stress | 0.2 MPa |
| Glass transition temp | 79–80 °C |
| Specific heat | 1767 J/kg °C |
| Conductivity | 0.13 W/m °C |
| density | 1.3253 kg/dm$^3$ |
| Shrinkage | 0.60% |

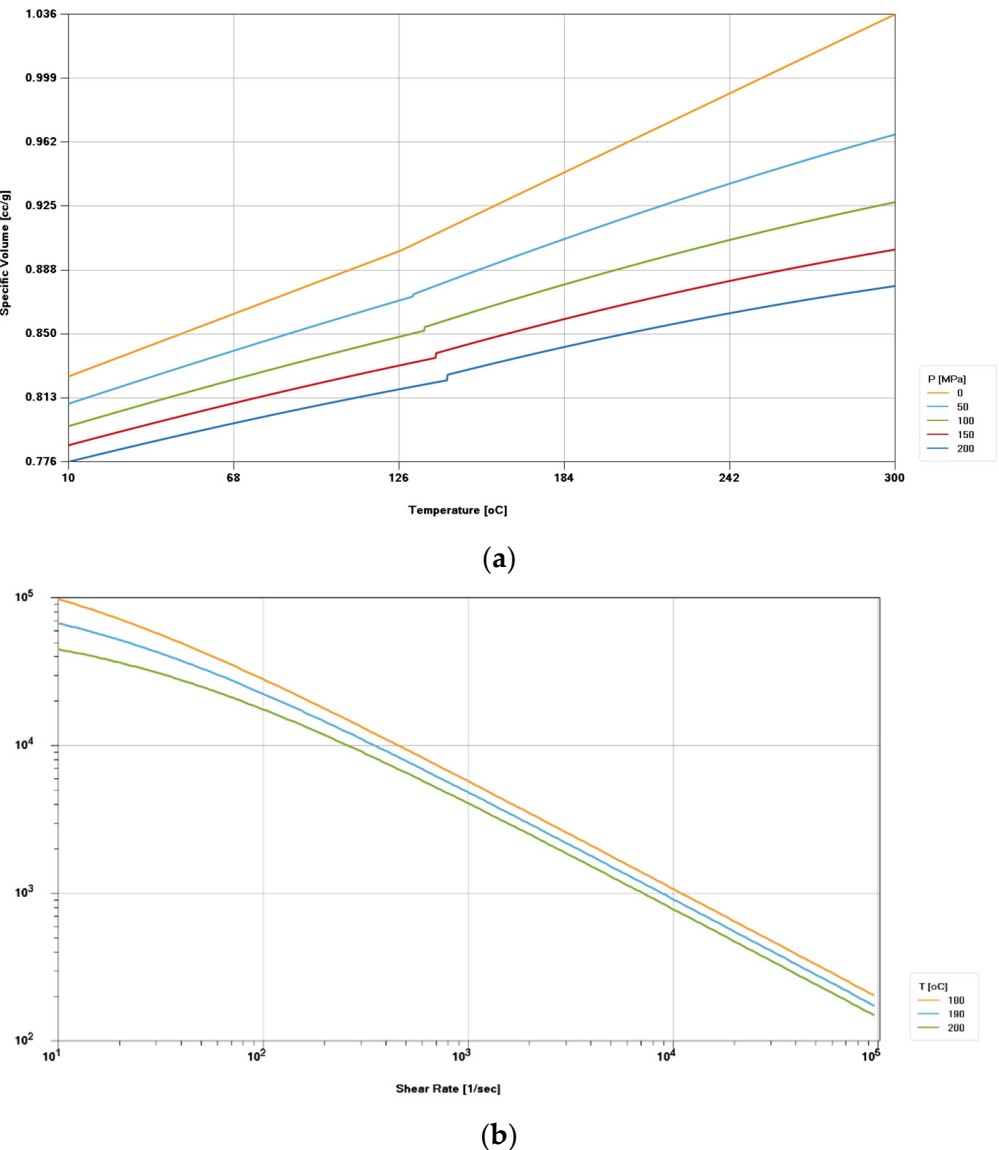

**Figure 2.** (**a**) pvT data and (**b**) viscosity curves of the Benvic IR705 PVC material.

As the first step, the CAD model of the bushing was created. This model was imported into the Autodesk Moldflow software for mesh generation and creating the runner system. Concurrently, the bushing has been masked using 459,096 3D tetragonal elements for both mold cavities. The overall element size of the bushing is 3.5 mm, and due to the greater sensitivity of the gate location area, the element size of this area is set to 2 mm.

His section may be divided by subheadings. It should provide a concise and precise description of the experimental results, their interpretation, as well as the experimental conclusions that can be drawn.

### 2.2. Measuring the Simulation Results

A visualization of blush simulation is shown in Figure 3a. As an assumption, the shape of the defect was assumed, due to the visual similarity and to avoid adding unnecessary complexity to the problem, assumed to be similar to an ellipse. Therefore, the formula for calculating the area of the ellipse has been applied to determine the area of the defect. Figure 3b represents the measures needed to calculate of the defect area.

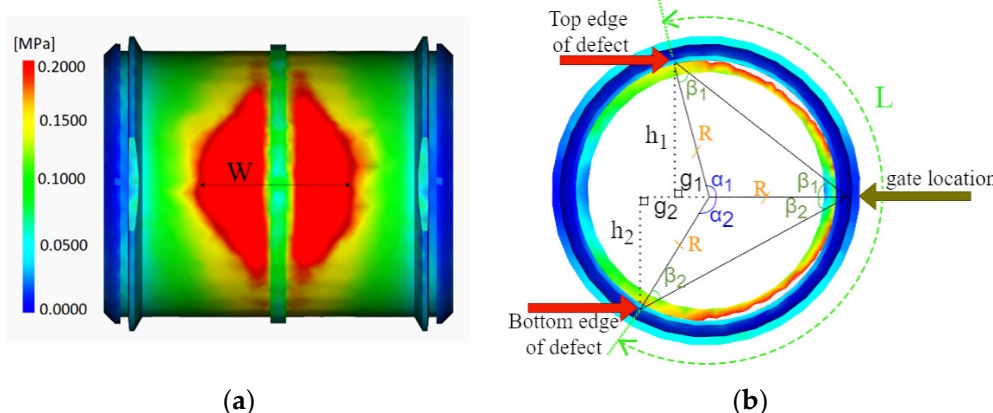

**Figure 3.** Width of defect on a simulated bushing (**a**) and geometrical measures for calculation of L (**b**).

The area of the ellipse can be calculated by Equation (1):

$$Area = \pi \times \frac{L}{2} \times \frac{W}{2} \tag{1}$$

where $L$ and $W$ represent the large and the small diameter of the ellipse, respectively. To calculate the area of the defect, a direct measurement of the value of $W$, the value of $L$ was calculated by measuring the length of the arc from top edge to bottom edge of the defect, as shown in Figure 3b, via Equations (2)–(7).

$$\beta_1 = \tan^{-1}(h_1 / g_1) \tag{2}$$

$$\alpha_1 = 180 - (2 \times \beta_1) \tag{3}$$

Having Equations (2) and (3), the angle of the top arc of defect ($\alpha_1$) can be calculated. Additionally, Equations (3) and (4) calculate the same parameter ($\alpha_2$) for the bottom part of the defect.

$$\beta_2 = \tan^{-1}(h_2 / g_2) \tag{4}$$

$$\alpha_2 = 180 - (2 \times \beta_2) \tag{5}$$

Equation (6) sums up the amounts of top and bottom angle of the defect's arc. Thus, $\alpha_T$ represents the total angle of the defect's arc.

$$\alpha_T = \alpha_1 + \alpha_2 \tag{6}$$

$$L = \frac{\alpha_T}{360} \times 2\pi R \tag{7}$$

According to Equation (7), the value of $L$ can be calculated. Having the values of $L$ and $W$, the area of the defect was calculated in each case through Equation (1).

### 2.3. Experimental Results and Measurements

To validate the simulations, experimental tests were carried out according to the procedure shown in Figure 4. In the first step, the PVC compound with the properties given in Table 1 was fed into the injection molding machine. Next, process parameters were set in the digital process setting panel. After that, the process started with closing the mold, and after injection, the bushing was manufactured. As the next step, accurate and high-resolution imaging was employed to obtain a high-quality representation of the bushing. Eventually, the images were processed by a computer software to better visualize the blush defect. For experimental measurement of the blush defect area, the length and

the width of the defect were measured. These measurements enabled us to calculate the defect's area using the ellipse area measurement formula (Equation (1)).

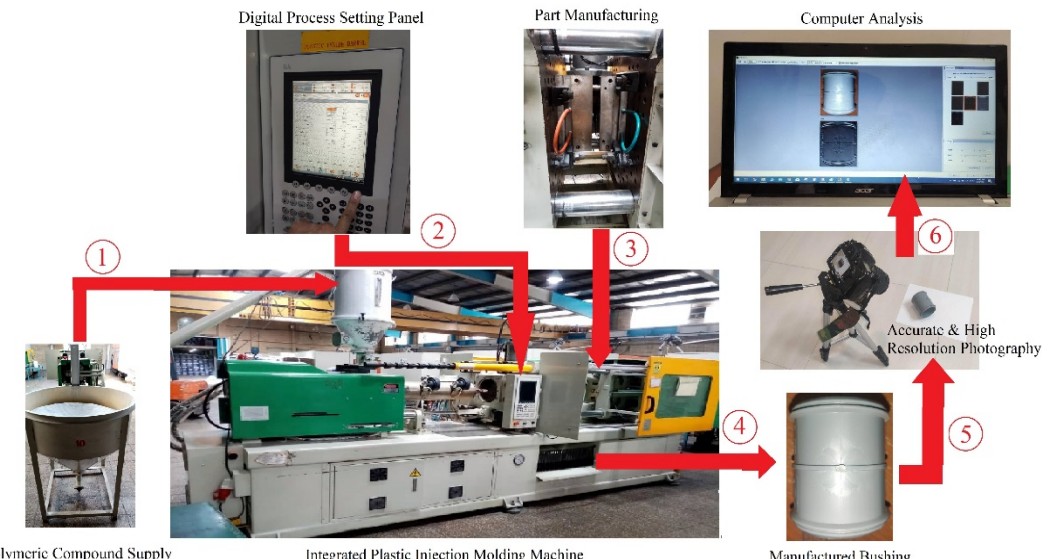

**Figure 4.** Experimental test procedure.

### 2.4. Function Approximation by ANOVA

Using Minitab software, a fractional factorial design was first created as the screening step, based on the eight previously mentioned parameters with a ratio of an 1/8. This ratio means that the number of experiments performed in the fractional DOE is 1/8 of the full factorial design. The number of full factorial DOE trials is 256, but with an 1/8 ratio, the fractional DOE is reduced to 32 experiments. The input parameters have been considered to all vary between two levels (see Table 2). After identifying the most effective parameters, a CCD design of the experiments will be conducted to clarify the impact of each effective parameter. After that, the results of ANOVA will represent the effective parameters and eventually a regression predictive model presented by the software.

**Table 2.** Levels of parameters required for performing simulation analyses.

| | Flow Rate (cm³/s) | Melt Temperature (°C) | Mold Temperature (°C) | Holding Pressure (MPa) | Gate Diameter (mm) | Runner Diameter (mm) | Gate Angle (°) | Included Angle (°) |
|---|---|---|---|---|---|---|---|---|
| Lower bound | 15 | 185 | 25 | 60 | 2.5 | 7 | 0 | 25 |
| Higher bound | 25 | 195 | 35 | 80 | 4.5 | 10 | 45 | 45 |

### 2.5. Basic and Hybrid Machine Learning Algorithms

#### 2.5.1. ANN

To provide a model for predicting the area of blush, several machine learning algorithms were considered and employed. The algorithms used in this research include basic ANN, combination of ANN and PSO, and combination of ANN and GA. The most straightforward ANN involves an input layer, a hidden layer, and an output layer. Each layer can contain one or more neurons. According to Equation (8), the value of each neuron is multiplied by the specified weight assigned to each link and added to the same value for all the other neurons in the same layer, and eventually enters the neurons of the next layer [7].

$$S_j = \sum_{j=0}^{N} x_i \times w_{ij} \qquad (8)$$

where $x_i$ is the output of the *ith* neuron of the previous layer, and $w_{ij}$ is the weight of the link between the *ith* neuron of the previous layer and the *jth* neuron of the present layer. $S_j$ represents the sum of the previous layer's outputs multiplied by the connection weights, which is the net input entering the $j^{th}$ neuron, and $N$ represents the number of inputs to the *jth* neuron in the hidden layer. Each neuron in the network produces its output ($O_j$) by entering $S_j$ that is a Tansig activator function similar to one indicated in Equation (9) [9]:

$$O_j = F(S_j) = \frac{1 - e^{-S_j}}{1 + e^{-S_j}} \tag{9}$$

### 2.5.2. Training ANN

- Basic ANN

In this research, the training was conducted with 70% of the available data provided in the DOE phase. Then, 15% of the available data was used as a validation dataset and the rest, 15% as a test dataset. In the first step, the weights were imported to the network randomly. Applying the gradient descent method causes these weights to be continuously updated during successive iterations. To train the network, the mean square error (*MSE*) of the predicted values should move towards minimization. Equations (10)–(12) represent this method [9,20]:

$$E = MSE = \frac{1}{N} \sum_{i=1}^{N} (y_p - y_t)^2 \tag{10}$$

$$\Delta w_{ij} = -\eta \frac{\partial E}{\partial w_{ij}} \times O_j \tag{11}$$

$$w_{ij}^m = w_{ij}^{m-1} + \Delta w_{ij} \tag{12}$$

where $\eta$ is the learning rate and controls the network convergence (a number between 0 and 1), $E$ represents the *MSE*, $N$ for the number of inputs, $y_t$ desired output, predicted output by the ANN, and $m$ indicates the iterations counter.

- ANN + GA

This method is similar to the basic ANN method, except that instead of using the gradient descend function used in basic ANN, the appropriateness of the weights assigned to each link is determined through the GA process, which is the selection of the best, the crossover, and the mutation, so that the MSE is minimized. In this algorithm, the probability of selecting each parent is assessed through Equations (13) and (14) [9]:

$$f_i = \frac{k}{m_i} \tag{13}$$

$$p_i = \frac{f_i}{\sum_{j=1}^{N} f_i} \tag{14}$$

where $k$ is a coefficient, $m_i$ symbolizes the fitness of $i_{th}$ input, $N$ substitutes for the number of generation population, and $j$ indicates the number of generations. Equation (15) is applied to combine the two chromosomes of $C_i$ and $C_h$ to produce the next generation [9].

$$\begin{cases} C_{ij} = C_{ij(1-b)} + C_{hjb} \\ C_{hj} = C_{hj(1-b)} + C_{ijb} \end{cases} \tag{15}$$

where $b$ substitutes for a random number between 0 and 1 and indicates the chromosome's intersection point. In addition, the Equations (16) and (17) are used to model the mutation in this algorithm [9].

$$C_{mn} = \begin{cases} C_{mn} + (C_{mn} - C_{\max}) \times f(g), r > 0.5 \\ C_{mn} + (C_{\min} - C_{mn}) \times f(g), r \leq 0.5 \end{cases} \tag{16}$$

$$f(g) = r_2(1 - {}^g\!/_{G_{\max}})\tag{17}$$

where $C_{mn}$ represents the gene $C_{mn}$, $C_{\min}$ and $C_{\max}$ stands for the higher and lower bounds for genes, and $r$ and $r_2$ are random numbers while $r$ is between 0 and 1, $g$ indicates the number of present generation population and $G_{max}$ is the maximum generations considered to iterate. So, the algorithm finally reaches a generation with perfectly fitting responses. A flowchart of the ANN + GA is shown in Figure 5 and its pseudocode is reported in Appendix A.

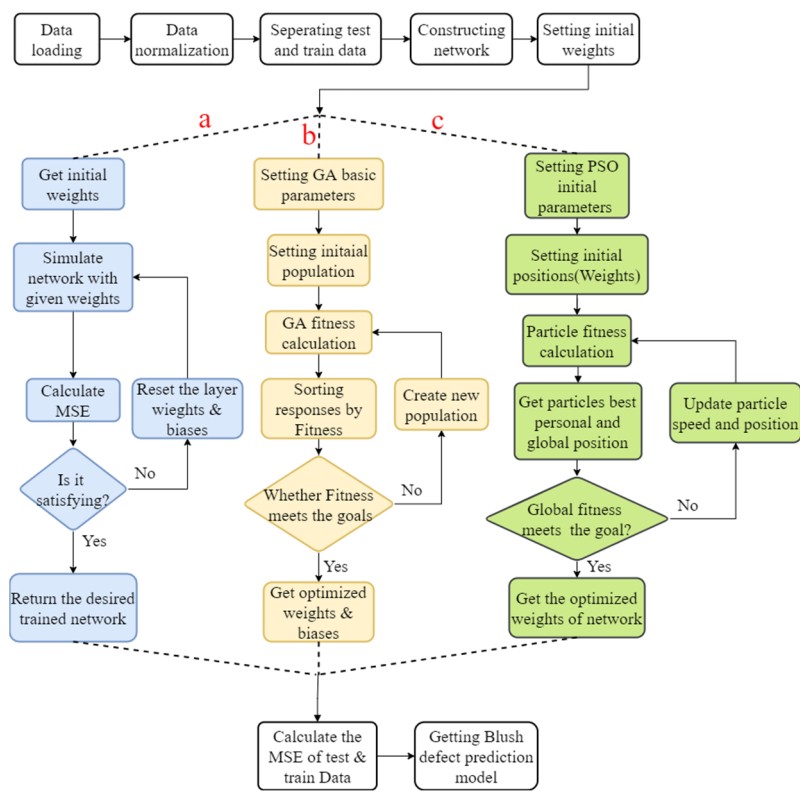

**Figure 5.** Training flowchart of: (a): basic ANN, (b): ANN + GA, (c): ANN + PSO.

- ANN + PSO

The PSO algorithm is a method established on swarm intelligence inspired by the behavior of a bird flock. The behavior of the particles, their speed, and the direction of their movement are influenced by the best experience of the particle itself in matching the goal and the best experience among all members of the population. The objective of this algorithm is to reduce the average fitness of all members of the population. In Equations (18)–(20), the best personal position of the particle is presented as *Pbest*, and the best position of the whole particle swarm is symbolized by *Gbest*. If the particles are located in an N-dimensional space, the vector $X_i = (X_{i1}, X_{i2}, \ldots, X_{iN})$ and the vector $V_i = (V_{i1}, V_{i2}, \ldots, V_{iN})$ represent the position and velocity of the particle $i$, respectively. In each generation, the values of these vectors must be updated and compared to the previous generation. The Equations (19)–(21)are used to update these values.

$$w(t) = w_{\max} - \frac{t(w_{\max} - w_{\min})}{t_{\max}}\tag{18}$$

$$v_{i,n}^{t+1} = wv_{i,n}^t + c_1 r_{1,n}(Pbest_{i,n}^t - x_{i,n}^t) + c_2 r_{2,n}(Gbest_n^t - x_{i,n}^t)\tag{19}$$

$$x_{i,n}^{t+1} = x_{i,n}^t + v_{i,n}^{t+1}\tag{20}$$

where $t$ is counter index for generations, $c_1$ and $c_2$ stand for two positive coefficients for acceleration while $r_{1,n}$ and $r_{2,n}$ are two random coefficients with uniform distribution in *Nth* dimension of the space, $n$ shows particle number, and $m$ represents the inertia weight employed [7]. Figure 5 depicts a flowchart of steps of optimizing layer weights and training the network using basic ANN, ANN + GA, and ANN + PSO. The pseudocodes for each of the three algorithms are also given in the Appendix A.

In Figure 5, the (a) route shows the basic ANN's flowchart, the (b) route shows the steps of ANN + GA, and the (c) represents the flowchart of ANN + PSO. To calculate the training accuracy of the networks, the Equations (21)–(23) are employed.

$$P_E = \frac{B_P - B_M}{B_M} \tag{21}$$

$$M_E = \frac{\sum\limits_{1}^{n} \left| P_{E_n} \right|}{n} \tag{22}$$

$$Accuracy_{Tr} = (1 - M_E) \times 100 \tag{23}$$

where $P_E$ indicates the prediction error for each data group, $B_P$ and $B_M$ represent the predicted value of the blush defect and the measured value of the defect, respectively. $n$ indicates the number of data sets, $M_E$ represents the average error of the whole data set, and $Accuracy_{Tr}$ shows the accuracy of training.

After training the networks, an optimization was conducted. Additionally, after training an accurate ANN, the prediction accuracy of the network has been measured, and the most accurate network has been employed as the fitness function of a GA to optimize the levels of design parameters. Then, the levels optimized by ANOVA and the GA will be compared together. In the flowchart shown in Figure 6, the research method has been shown. In this figure, the simulation and DOE sections indicate all the steps taken through the creation of digital twin, getting the results of it, and employing statistical analysis on the data. The ANN section demonstrates the steps of using different machine learning methods to analyze the data. Then, as the last step, the GA optimization section reveals the work that has been conducted to optimize the parameters' levels to reach the robust process setting with the lowest blush defect area.

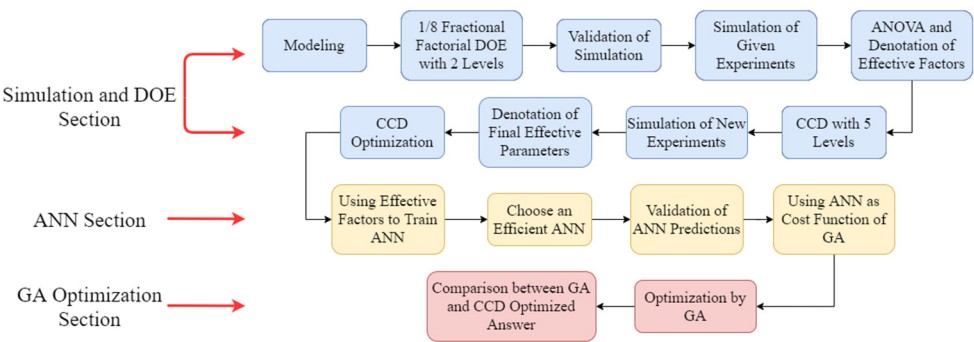

**Figure 6.** Flowchart of the research method.

## 3. Results and Discussion

### 3.1. FEA Validation

To ensure the accuracy of the FEA, experimental tests were performed and compared with the simulation results. Table 3 shows the validation results. As can be observed from this table, the results of FEA are in relatively good agreement with the results of the experimental tests, and the absolute value of the average deviation error is 5.6%. The parameters of flow rate, melt temperature, and holding pressure were varied to verify the

results. Other parameters remained constant due to practical limitations in changing the geometry of the mold.

**Table 3.** Validation of FEA.

|  | Experimental Test No. | 1 | 2 | 3 | 4 |
|---|---|---|---|---|---|
| **Inputs** | **Flow rate (cm³/s)** | 15 | 15 | 25 | 25 |
| | **Melt temperature (°C)** | 185 | 195 | 185 | 195 |
| | **Mold temperature (°C)** | 35 | 35 | 35 | 35 |
| | **Holding pressure (MPa)** | 60 | 80 | 60 | 80 |
| | **Runner diameter (mm)** | 10 | 10 | 10 | 10 |
| | **Gate diameter (mm)** | 3.5 | 3.5 | 3.5 | 3.5 |
| | **Gate angle (°)** | 0 | 0 | 0 | 0 |
| | **Included angle (°)** | 45 | 45 | 45 | 45 |
| **Outputs** | **Experimental defect area (mm²)** | 2108 | 1621 | 2483 | 1696 |
| | **Simulation defect area (mm²)** | 2212 | 1553 | 2777 | 1674 |
| | **Deviation error** | 4.9% | −4.2% | 11.8% | −1.3% |

According to the shear stress heat map, the area that exceeded the maximum allowable shear stress (0.2 MPa) appeared as the blush [34]. As can be concluded from Table 3, lower amounts of error were observed for simulations associated with process configurations, resulting in a smaller area of defect. Since the objective of the study is to decrease the area of defect, the closer the study gets to its goal, the lower the error amount. Figure 7 shows a comparison between the simulation results for maximum shear stress and the blush defect in an experimental test performed with the same parameter settings.

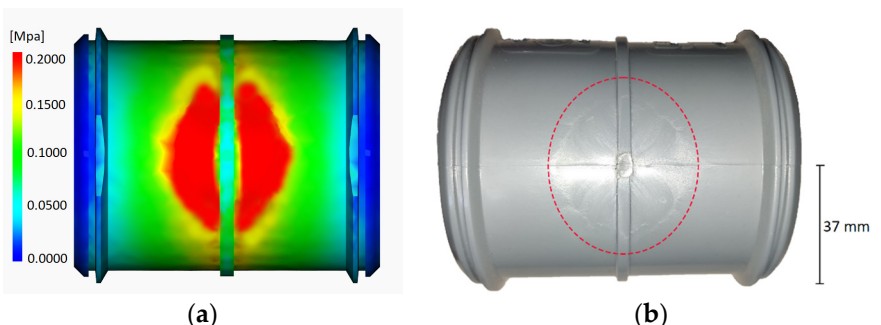

(**a**)　　　　　　　　　　　　　　　　　(**b**)

**Figure 7.** Simulation result (**a**) and enclosed area of defect in the experimental test (**b**) obtained with the same injection molding process settings (melt temperature = 185 °C, mold temperature = 35 °C, flow rate = 25 cm³/s, holding pressure = 80 MPa, runner diameter = 10 mm, gate diameter = 3.5 mm, gate angle = 0°, and included angle = 45°).

### 3.2. Statistical Analysis

To determine the effective parameters, a screening step was implemented using the fractional factorial design. The ANOVA results are given in Table 4. These results indicate that the parameters of flow rate, melt temperature, holding pressure, and runner diameter were effective due to the lower *p*-value of 0.05. The parameters of mold temperature, gate diameter, gate angle, and included angle only slightly impacted the defect area. Furthermore, the meaning of squares, which is obtained by dividing the treatment sum of squares by the degrees of freedom, can be used to determine which factors are significant. The higher the mean square, the more of an effect it has on the results [38,39].

**Table 4.** ANOVA results of the screening step.

| No. | Source | Mean of Squares | *p*-Value |
|---|---|---|---|
| 1 | Model | 3,424,612 | 0.011 |
| 2 | Flow rate | 4,978,362 | 0.011 |
| 3 | Melt temperature | 19,348,900 | 0.001 |
| 4 | Mold temperature | 870,044 | 0.138 |
| 5 | Holding pressure | 6,600,269 | 0.007 |
| 6 | Runner diameter | 32,488,905 | 0.000 |
| 7 | Gate diameter | 458,198 | 0.250 |
| 8 | Gate angle | 1946 | 0.934 |
| 9 | Included angle | 96,108 | 0.571 |

The DOE intended for the screening step to be executed with all parameters varying between two levels. In the second phase, by performing CCD with an alpha coefficient of 1.4, the number of parameter levels was increased to five (shown in Table 5). This could lead to a better understanding of the parameters' effects on the process outputs.

**Table 5.** The levels of CCD.

| No. | Flow Rate (cm$^3$/s) | Melt Temperature (°C) | Holding Pressure (MPa) | Runner Diameter (mm) |
|---|---|---|---|---|
| 1 | 12 | 185.2 | 54.6 | 6.4 |
| 2 | 14 | 187 | 59 | 7 |
| 3 | 19 | 191 | 70 | 8.5 |
| 4 | 24 | 196 | 81 | 10 |
| 5 | 26 | 197.8 | 85.4 | 10.6 |

To predict the area of the defect with a regression equation, an ANOVA was implemented on the results of the simulations suggested by the CCD. According to Table 6 and considering the *p*-value of the model equal to 0.000, it can be concluded that the model presented by ANOVA has good accuracy in the prediction of the defect area. The effect of all four parameters known to be effective in the previous step has been reaffirmed by the design of experiments carried out by considering more levels. Diagrams related to effective parameters can be seen in Figure 8.

**Table 6.** Results of the second ANOVA (CCD).

| No. | Source | Mean of Squares | *p*-Value |
|---|---|---|---|
| 1 | Model | 1,463,290 | 0 |
| **2** | **Linear parameters** | **4,473,581** | **0** |
| 3 | Flow rate | 1,209,310 | 0.039 |
| 4 | Melt temperature | 1,369,223 | 0.049 |
| 5 | Holding pressure | 460,154 | 0.027 |
| 6 | Runner diameter | 16,845,635 | 0 |
| **7** | **Squares** | **533,923** | **0.008** |
| 8 | Flow rate × Flow rate | 6013 | 0.815 |
| 9 | Melt temperature × Melt temperature | 53,785 | 0.486 |
| 10 | Holding pressure × Holding pressure | 527 | 0.945 |
| 11 | Runner diameter × Runner diameter | 1,474,471 | 0.002 |
| **12** | **Interactions** | **76,008** | **0.641** |
| 13 | Melt temperature × Flow rate | 4789 | 0.834 |
| 14 | Melt temperature × Holding pressure | 9658 | 0.767 |
| 15 | Melt temperature × Runner diameter | 50,917 | 0.498 |
| 16 | Flow rate × Holding pressure | 126 | 0.973 |
| 17 | Flow rate × Runner diameter | 175,496 | 0.216 |
| 18 | Holding pressure × Runner diameter | 215,063 | 0.173 |

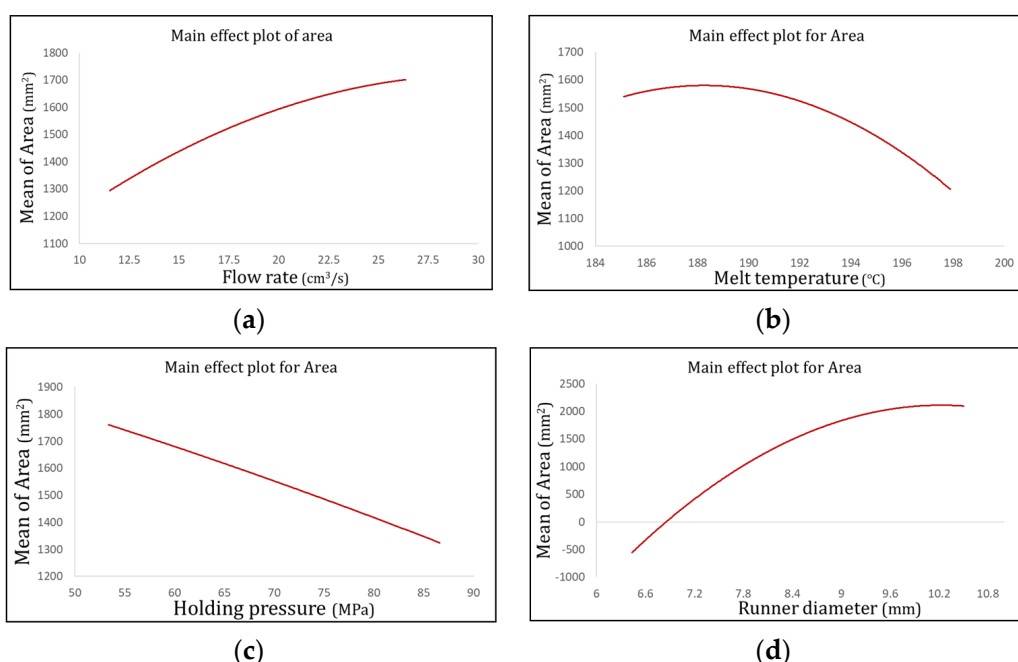

**Figure 8.** The effect of: (**a**) flow rate, (**b**) melt temperature, (**c**) holding pressure, and (**d**) runner diameter on the defect area.

Figure 8 illustrates the manner of the impact of each parameter on the results. According to the *p*-value obtained from the effects of ANOVA, the runner diameter has the most significant impact on the area of the blush defect. After that, the parameters of holding pressure, flow rate, and melt temperature have the highest impact on the defect area, respectively. Utilizing ANOVA, a regression equation to predict the area of the defect has been worked out using all input parameters presented in Equation (24).

$$Area = -12367 + (25.7 \times A) - (25.4 \times B) - (13.82 \times C) + (3898 \times D) - (193.2 \times D \times D) \qquad (24)$$

where *A* indicates the flow rate, *B* the melt temperature, *C* the holding pressure, and *D* represents the runner diameter. The standardized residual and histogram plot shown in Figure 9 reveal the accuracy of Equation (24). As can be seen from the Versus Order plot, the predicted data are in general equally scattered around the zero line. Additionally, the histogram plot in this figure shows that the amount of data with a standard residual equal to or near zero is much higher than the amount of data with higher residuals. Thus, it can be concluded that the defect area amounts predicted by the regression model are in good agreement with the FEA results.

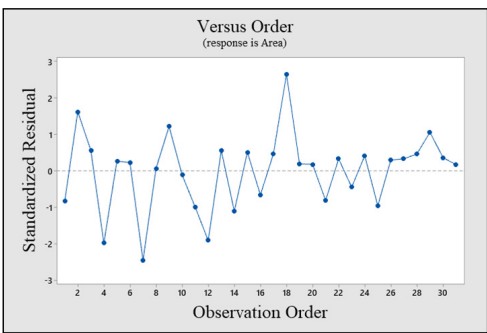
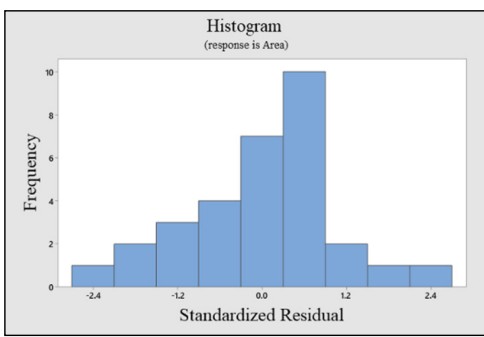

**Figure 9.** Standardized residual and histogram graph.

### 3.3. ANN Validation and Comparison with ANOVA

After training the ANN, the network should be verified with the data intended for testing. The test data in this study included 15% of the total data that had not been used in the ANN training phase. Table 7 also compares the prediction accuracy of different types of ANN with that presented by ANOVA.

**Table 7.** Comparison of response prediction methods.

| No. | Predictor | Neurons in Hidden Layer | Number of Particles | Population Size | Iterations | Accuracy$_{Tr}$ | Training Time (s) |
|---|---|---|---|---|---|---|---|
| 1 | | 4 | | | 178 | 99.96% | 2 |
| 2 | Basic ANN | 6 | | | 134 | 99.97% | 2 |
| 3 | | 8 | | | 91 | 99.99% | 1 |
| 4 | | 10 | | | 102 | 99.98% | 1 |
| 5 | | | 10 | | 245 | 97.88% | 59 |
| 6 | ANN + PSO | | 30 | | 198 | 98.23% | 178 |
| 7 | | | 50 | | 179 | 98.78% | 264 |
| 8 | | | 70 | | 171 | 99.26% | 408 |
| 9 | | | | 10 | 276 | 98.76% | 51 |
| 10 | ANN + GA | | | 30 | 251 | 98.61% | 159 |
| 11 | | | | 50 | 237 | 98.33% | 256 |
| 12 | | | | 70 | 209 | 98.17% | 357 |
| 13 | ANOVA | | | | | 86.57% | - |

According to Table 7, the best networks of each type have been chosen to compare the prediction accuracy. For basic ANN, the trained network in row 3 is considered as the selected basic ANN with the best performance. The type of activator function is "TanSig" for the first layer and "Linear" for the second layer. For ANN + PSO, the trained network in row 7 is regarded as the most suitable algorithm of this type. The trained network in row 11 has also been chosen as the most suitable ANN + GA combination. The mutation rate in this algorithm is 15%, and the cross-over rate is 65%. Additionally, Figure 10 exhibits a comparison between the predicted values via the regression formula provided by ANOVA, selected neural networks of all three types (basic ANN, ANN + GA, and ANN + PSO) with the value obtained from the FEA. Furthermore, Table 8 presents the values predicted by each algorithm for all the data sets compared with ANOVA predicted values and FEA results.

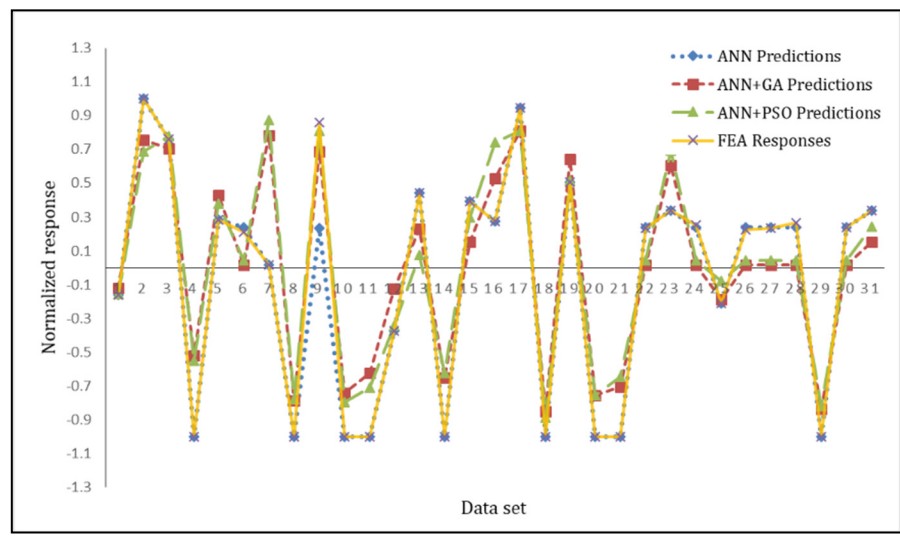

**Figure 10.** Comparison of normalized neural network and ANOVA prediction values with FEA results.

**Table 8.** Comparison of normalized values predicted by ANOVA, ANN Algorithms, and FEA.

| No. | FEA | ANOVA | Basic ANN | ANN + PSO | ANN + GA |
|---|---|---|---|---|---|
| 1 | −0.161 | 0.298 | −0.161 | −0.157 | −0.119 |
| 2 | 1 | 0.849 | 1 | 0.688 | 0.754 |
| 3 | 0.762 | 0.83 | 0.762 | 0.779 | 0.704 |
| 4 | −1 | −0.219 | −1 | −0.551 | −0.52 |
| 5 | 0.284 | 0.477 | 0.284 | 0.38 | 0.43 |
| 6 | 0.21 | 0.417 | 0.237 | 0.044 | 0.016 |
| 7 | 0.015 | 0.706 | 0.015 | 0.876 | 0.783 |
| 8 | −1 | −0.571 | −1 | −0.781 | −0.786 |
| 9 | 0.859 | 0.799 | 0.233 | 0.811 | 0.686 |
| 10 | −1 | −0.54 | −1 | −0.797 | −0.745 |
| 11 | −1 | −0.389 | −1 | −0.71 | −0.625 |
| 12 | −0.376 | 0.311 | −0.376 | −0.342 | −0.124 |
| 13 | 0.444 | 0.558 | 0.444 | 0.074 | 0.231 |
| 14 | −1 | −0.37 | −1 | −0.623 | −0.65 |
| 15 | 0.393 | 0.523 | 0.393 | 0.296 | 0.152 |
| 16 | 0.274 | 0.629 | 0.274 | 0.742 | 0.531 |
| 17 | 0.945 | 1 | 0.945 | 0.809 | 0.81 |
| 18 | −1 | −1 | −1 | −0.886 | −0.854 |
| 19 | 0.507 | 0.679 | 0.507 | 0.517 | 0.644 |
| 20 | −1 | −0.59 | −1 | −0.757 | −0.755 |
| 21 | −1 | −0.42 | −1 | −0.648 | −0.705 |
| 22 | 0.233 | 0.417 | 0.237 | 0.044 | 0.016 |
| 23 | 0.34 | 0.647 | 0.34 | 0.693 | 0.606 |
| 24 | 0.252 | 0.417 | 0.237 | 0.044 | 0.016 |
| 25 | −0.214 | 0.276 | −0.214 | −0.081 | −0.191 |
| 26 | 0.225 | 0.417 | 0.237 | 0.044 | 0.016 |
| 27 | 0.233 | 0.417 | 0.237 | 0.044 | 0.016 |
| 28 | 0.263 | 0.417 | 0.237 | 0.044 | 0.016 |
| 29 | −1 | −0.741 | −1 | −0.82 | −0.837 |
| 30 | 0.24 | 0.417 | 0.237 | 0.044 | 0.016 |
| 31 | 0.336 | 0.536 | 0.336 | 0.241 | 0.151 |

As it is clear from Figure 10 and Table 8, ANOVA with 86.5% accuracy and basic ANN with 99.99% accuracy provide the farthest and the closest predictions to the FEA response, respectively. Additionally, according to these data, the basic ANN presented in row 3 of Table 7 (with eight neurons in the hidden layer) has been chosen as the most appropriate prediction model. The graph presented in Figure 11 indicates the training procedure of the basic ANN. During 91 epochs, the basic ANN converged. The MSE trend of training data can be seen alongside this trend for validation and test data. In addition, this figure represents the regression analysis of the basic ANN model for training, validation, and test data. Additionally, the regression model for the whole data set can be seen in the bottom right corner of Figure 11.

### 3.4. Optimization Using GA

To optimize the parameter levels, a trained ANN with the specifications listed in the third row of Table 4 is used as the cost function. The parameters of melt temperature, flow rate, holding pressure, and runner diameter are considered as the inputs of the GA, and the area of the blush defect is regarded as the only output. The allowable range for each of these parameters is specified in Table 3. Additionally, for the GA, values of 300 were taken into account as the initial population, 300 as the maximum number of generations, 60% as the crossover rate, and 40% as the mutation rate. During the optimization process, the value of the cost function of the GA is constantly decreasing, which indicates the proper performance of the algorithm in finding the optimal values. At the end of the process, the algorithm had reached the optimal response after approximately 170 iterations, after which the response showed little change. The algorithm introduces the values of 197.1,

25.6, 84.9, and 6.4 as the optimal levels for the parameters of melt temperature, flow rate, holding pressure, and runner diameter, respectively. Figure 12 shows the optimization process by the GA. Using experimental validation, it is possible to compare the results of the experimental tests, GA, and ANOVA. This comparison can be seen in Table 5.

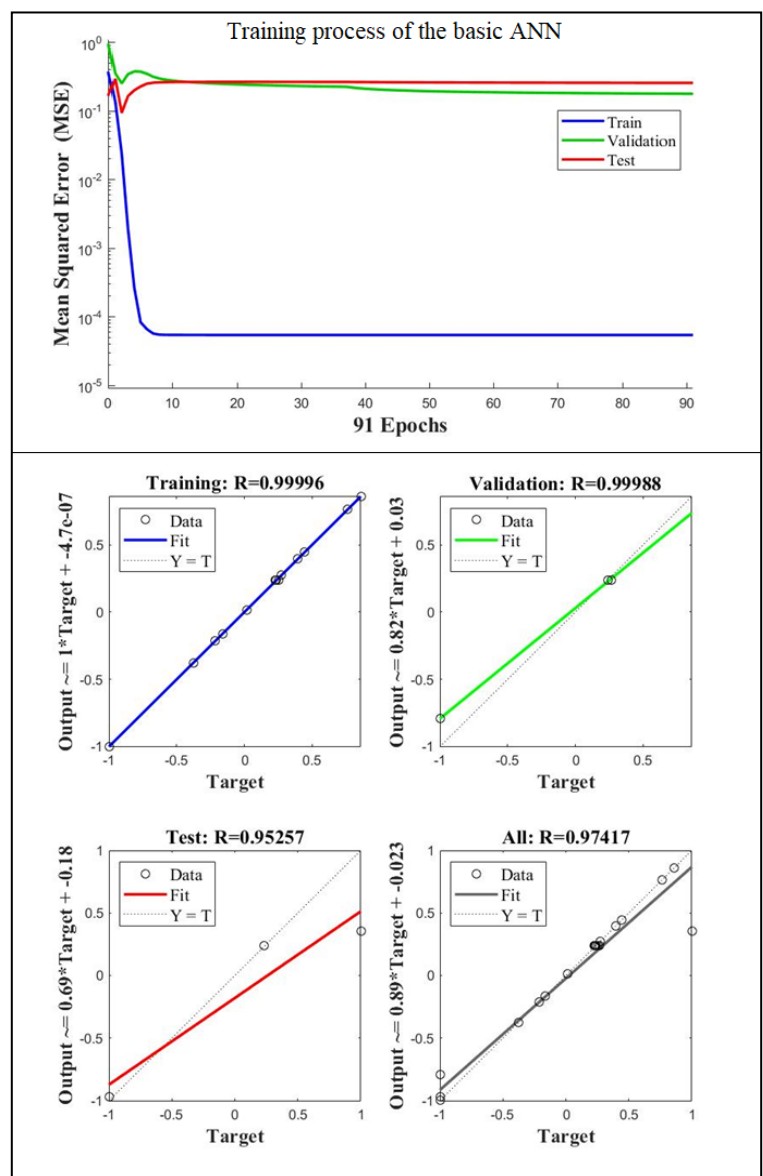

**Figure 11.** Regression analysis of the basic ANN model.

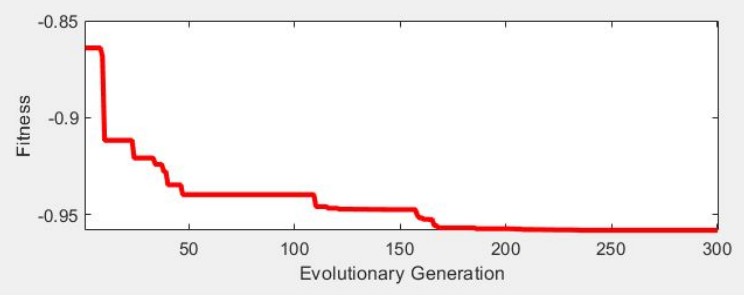

**Figure 12.** Optimization process by GA.

From the results obtained from Table 9, it can be seen that, on account of optimization using GA, the amount of blush defect area has been reduced by 81.7%. Additionally, the optimization performed using the CCD method has reduced the defect area by 74.0%. Figure 13 shows a visual comparison between the blush area, before and after optimization in the injection molding simulation, alongside the image of the defect in the bushing produced with the optimized parameters.

**Table 9.** Comparison of the defect area before and after optimization.

| Title | Melt Temperature (°C) | Flow Rate (cm³/s) | Holding Pressure (mm) | Runner Diameter (mm) | Defect Area (mm²) |
|---|---|---|---|---|---|
| Initial bushing | 197.0 | 12.0 | 35.0 | 10.0 | 1978 |
| Optimal bushing suggested by CCD optimization | 197.8 | 26.0 | 56.4 | 6.7 | 517 |
| Optimal bushing suggested by GA | 197.1 | 25.6 | 84.9 | 6.4 | 362 |
| Experimental test on the dataset suggested by GA | 197.1 | 25.6 | 84.9 | 6.4 | 366 |

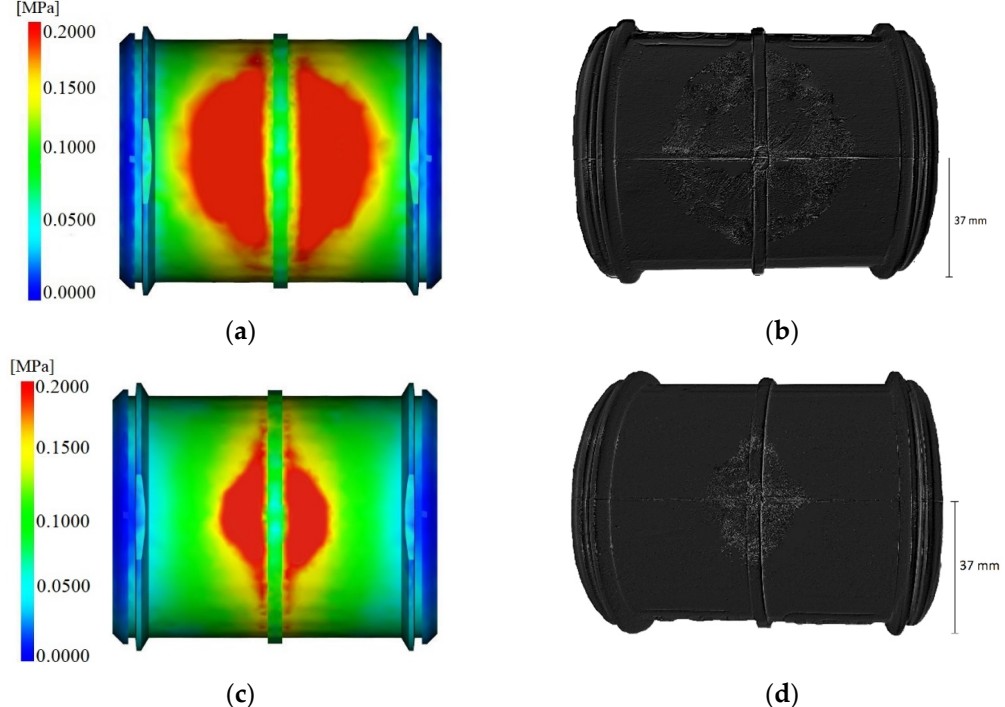

**Figure 13.** The initial defected bushing area is in simulation (**a**) and experimental test (**b**); and the bushing is after optimization in simulation (**c**) and experimental test (**d**).

## 4. Conclusions

This study aimed at the determination of the effect of eight process parameters (flow rate, melt temperature, mold temperature, holding pressure, runner diameter, gate diameter, gate angle, and included angle) on the blush defect in PVC bushings produced by injection molding. Some prediction models have been created to estimate the area of the blush defect using ANOVA and ANNs. Among the prediction models used, the basic ANN method with a training accuracy of 99.99% has shown the best performance compared to other prediction methods for predicting the FEA results.

Results showed that flow rate, melt temperature, and runner diameter have a particularly strong effect on the blush defect and can be related to the viscosity of the molten material. The viscosity can increase with rapid cooling of the material, making the material's shear stress exceed the allowable limit and, in turn, promoting blush.

Holding pressure has also affected the blush defect. With the increase in holding pressure, the blush defect decreases. Lower values of holding pressure can cause some semi-cooled material (which has a high viscosity) to enter the mold cavity. Reflowing the material in these high viscosity conditions can cause high amounts of shear rate, which is the underlying reason of blush defect occurrence. The parameters of mold temperature, gate diameter, gate angle, and included angle have a negligible effect on the result.

The key results of the research can be summarized as follows:

- After optimization using the GA through the trained ANN as the cost function, values of 197.1 °C, 25.6 cm$^3$/s, 84.9 MPa, and 6.4 mm were introduced as the optimal levels for the parameters of melt temperature, injection flow rate, holding pressure, and runner diameter, respectively.
- After applying the optimized parameter levels by GA in a new FEA, the defect area was reduced by 81.7% compared to the pre-optimization defect area.

**Author Contributions:** Methodology, A.M.A. and G.T.; Software, A.M.A.; Validation, A.M.A., R.S. and G.T.; Formal analysis, R.S.; Investigation, A.M.A.; Resources, G.A.; Writing—original draft, A.M.A.; Writing—review & editing, G.A., Y.S., M.C., J.H.H., M.K. and G.T.; Visualization, A.M.A.; Supervision, G.A. and G.T.; Project administration, G.A. All authors have read and agreed to the published version of the manuscript.

**Funding:** This research work was partially supported by the DIGIMAN4.0 project ("DIGItal MANufacturing Technologies for Zero-defect Industry 4.0 Production", https://www.digiman4-0.mek.dtu.dk/, accessed on 13 February 2023). DIGIMAN4.0 is a European Training Network supported by Horizon 2020, the EU Framework Programme for Research and Innovation (Project ID: 814225).

**Institutional Review Board Statement:** Not applicable.

**Informed Consent Statement:** Not applicable.

**Data Availability Statement:** Not applicable.

**Acknowledgments:** This research work was undertaken in the context of DIGIMAN4.0 project ("DIGItal MANufacturing Technologies for Zero-defect Industry 4.0 Production", http://www.digiman4-0.mek.dtu.dk/). DIGIMAN4.0 is a European Training Network supported by Horizon 2020, the EU Framework Programme for Research and Innovation (Project ID: 814225). Yazd Poolica Co. (Yazd, Iran) is thanked for all their support during the experimental steps of the study.

**Conflicts of Interest:** The authors declare no conflict of interest.

## Appendix A. Pseudocodes of the Algorithms

- ANN
    1. Start of program
    2. Train percent = tr
    3. Test percent = ts
    4. Initialize data
    5. Separate input and output data
    6. Normalize all data
    7. Initialize a network structure
    8. Set a random matrix for initial layer weights and biases
    9. repeat
    10. Use tr percent of data for training network
    11. Use Equation (10) to Evaluate trained network with ts percent of data
    12. Use Equations (11) and (12) to reset the weights and biases
    13. Until termination criteria
    14. Simulate the trained network with input data
    15. Calculate and report the MSE

For the ANN + PSO and ANN + GA algorithms, all steps are similar to the above pseudocode. Only the steps for optimizing layer weights (lines 8–12) are overwritten. In other words, the following pseudocodes are replaced with the iterative loop in the upper pseudocode.

- ANN + PSO

  1. Start of weight optimization
  2. for each particle(each neural network)
  3. Initialize particle position(Network weights) and velocity vectors
  4. end for
  5. Fitness = f(X)i
  6. Personal best position = Xpb
  7. Global best position = Xgb
  8. repeat
  9. for particlei i = 1 to Nparticles
  10. Fitness = calculate the Fitness of particlei
  11. if f(Xbp)i < f(Xbp)
  12. Xbp = Xi, Pbest = Fitnessi
  13. end if
  14. if f(Xgb)i > f(Xgb)
  15. Xgb = Xi, Gbest = Fitnessi
  16. end if
  17. update the velocity of the particle using Equations (18 and (19)
  18. update the position of the particle using Equation (20)
  19. end for
  20. Until termination criteria

- ANN + GA

  1. Start of weight optimization
  2. N = number of network weights
  3. G = Number of maximum generations
  4. RecomPercent = r/100
  5. CrossPercent = c/100
  6. MutatPercent = 1 − RecomPercent − CrossPercent
  7. Initialize genes randomly(initial Network weights)
  8. Calculate fitness = f(X)i
  9. Sort Chromosomes according to fitness
  10. for i = 1 to G
  11. Create new population(RecomPercent × N + CrossPercent × N + MutatPercent × N)
  12. Calculate fitness
  13. Sort Chromosomes according to fitness
  14. end for

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
