# Peer review of "Application of Machine Learning for Prediction and Process Optimization—Case Study of Blush Defect in Plastic Injection Molding"

_applsci, doi:10.3390/app13042617_

Round 1
Reviewer 1 Report
The three first paragraphs of section 2 are copied from the template.
Is the cad model described in Fig. 1 ad-hoc or based-on literature, standards, or normative?
Line 103, check grammar.
I suggest expanding figure 3
The method to calculate the ellipse area (since the defect geometry is not exactly an ellipse) is not clear. This implies an a-prior error which should almost be considered/mentioned (it would feel like a gaussian curve instead an ellipse).
Define in line 254 all the ANN algorithms used (not only acronyms).
The authors indicate “to provide a model for predicting the blush” It is not clear the parameter to predict. The use of ANN (indicated in Fig 6) should be more justified in the text.
The methods to measure the parameters in the experimental work should be described. I believe that these measures will be taken automatically by the machine but it should be indicated (and accurate/resolution features if they are known).
Figure 7 b is not adequately described what is the additional information with respect a). Idem in Fig 13.
The relevant conclusions of ANOVA of Table 4 must be scientifically commented on when the table is shown (not only mentioned the p-value).
Appendix titled should be defined.
Author Response
Thanks for your time and consideration. I humbly added the file including the responses to your precious comments.
Best regards,
Alireza

Reviewer 2 Report
This manuscript presented the application of machine learning for the prediction of blush defect in a plastic injection molded part and the genetic algorithm was applied further to reduce the blush defect. The results demonstrated the effective method. However, the manuscript still needs to be modified.
1) The title should be modified. "Optimization of Blush Defect" is not suitable.
2) Abbreviations should be defined at first mention.
3)The relevant technical methods are listed in the Introduction section, but not classified.
4) Three paragraphs from the "template" should be deleted.
5) Some dimensions in Figure 1 are not in accordance with the values of gate diameter and runner diameter in Table 2.
6) Some wrong words and typos and the format, especially for the equations, should be revised.
7) It seems that 31 group data for CCD were used, please explain this.
8) The Conclusion section is too long.
Author Response
Thanks for your time and consideration. I humbly added the file including the responses to your precious comments.
Best regards

Reviewer 3 Report
Reviewer Comment
Manuscript Number: applsci-2203962
Dear Editor,
The Abstract and Introduction sections are well arranged and successfully described the objective and significance of the research.
Despite the effort of the authors, consideration of these points are vital to make this submitted article qualified for publication in this esteem journal.
1- ALL terms should be explained from the beginning of the submitted manuscript, even though for those they are famous, such as Finite Element Analysis (FEM), Artificial Neural Network (ANN), Response Surface Methodology (RSM), Design of Experiment (DOE), ...etc.
2-Line 63 and forward: Response Surface Methodology (RSM) instead of: RSM (Response Surface Methodology) !
Non-Dominated Sorting Genetic Algorithm (NSGA-II) is true
Particle Swarm Optimization (PSO) is true
and for other acronyms should follow the same abbreviation arrangement.
....etc
3- When use any term, e.g. FEM; it should be used throughout the manuscript consistently. No need to use Finite Element Analysis again in line 105. The same in consistence of using different terms was observed in several sections of the manuscript.
4-for the DOE, RSM was selected based on CCD. Which software was used to design the experiments?
5- for the same CCD, what is the value of alpha (α)
6-CCD has several designs, such as FCC, circumscribed and inscribed. Which one was used in the current study? and why this design was selected among others?
7- Section 2 Materials and Methods is an exact copy & paste from the Journal template ! The authors have write their own words.
8- Details of ANN, RSM (CCD), FEM , software used, taken ranges, levels, steps have been taken, and all other details should be described in the section 2.2.
9-In the section 3.2 and 3.3, several references to verify the significance of R2 should be added, and qualify any model based on how the value of R2 approaches 1.
https://doi.org/10.1016/j.sciaf.2022.e01282
https://doi.org/10.1016/j.ijrmhm.2018.03.006
https://doi.org/10.1016/j.chemosphere.2022.137665
https://doi.org/10.1016/j.jobe.2021.102788
10-for the data used from published articles, it should be presented in the Appendix section. The full data driven from these literature review can be arranged in Tables and presented clearly in appendixes.
11- Conclusion section is recommended to be preciser
Author Response
Dear reviewer,
Thanks for your time and consideration. I humbly added the file including the responses to your precious comments.
Best regards,

Round 2
Reviewer 1 Report
The limitation described in comment 5 (ellipse) must be clearly justified in the manuscript.
Author Response
Dear reviewer,
Thank you very much for your time and effort. I appreciate.
please find the answers to your comments in the attached file.
Best regards.

Reviewer 3 Report
Reviewer Comment
Manuscript Number: applsci-2203962
Dear Editor,
The authors corrected several aspects required to make this submitted article suitable for publication in your esteem journal. However, the conclusion section should be preciser, it is too long.
Author Response
Dear reviewer,
Thank you very much for your time and effort. I appreciate.
Please find the file of answers to your comments in the attachment.
Best regards.
